# Influence of the Process Parameters on the Properties of Cu-Cu Ultrasonic Welds

**Koen Faes [1,*], Rafael Nunes [1], Sylvia De Meester [2], Wim De Waele [2], Felice Rubino [3] and Pierpaolo Carlone [4]**

[1] Belgian Welding Institute, 9052 Zwijnaarde, Belgium
[2] Department of Electromechanical, Systems and Material Engineering, Ghent University, 9052 Ghent, Belgium
[3] Department of Civil and Mechanical Engineering, University of Cassino and Southern Lazio, 03043 Cassino, Italy
[4] Department of Industrial Engineering, University of Salerno, Via Giovanni Paolo II, 132, 84084 Fisciano, Italy
[*] Correspondence: koen.faes@bil-ibs.be; Tel.: +32-9-292-14-00

**Abstract:** Ultrasonic welding (USW) is a solid-state welding process based on the application of high frequency vibration energy to the workpiece to produce the internal friction between the faying surface and the local heat generation required to promote the joining. The short welding time and the low heat input, the absence of fumes, sparks or flames, and the automation capacity make it particularly interesting for several fields, such as electrical/electronic, automotive, aerospace, appliance, and medical products industries. The main problems that those industries have to face are related to the poor weld quality due the improper selection of weld parameters. In the present work, 0.3 mm thick copper sheets were joined by USW varying the welding time, pressure, and vibration amplitude. The influence of the process variables on the characteristics of the joints and weld strength is investigated by using the analysis of variance. The results of the present work indicate that welding time is the main factor affecting the energy absorbed during the welding, followed by the pressure and amplitude. The shear strength, on the other hand, resulted mostly influenced by the amplitude, while the other parameters have a limited effect. Regardless the welding configuration adopted, most welds registered a failure load higher than the base material pointing out the feasibility of the USW process to join copper sheets.

**Keywords:** ultrasonic welding; copper; parameter optimization

## 1. Introduction

Ultrasonic welding (USW) is a friction-based welding technology which is based on the phenomenon of interfacial friction, which joins metals by diffusion and the adhesion of softened metals [1,2]. USW is a solid-state welding process in which coalescence at the faying surfaces is produced by local application of high frequency vibrational energy, while the workpieces are held together a under moderately low static pressure [3]. This leads to local heat generation due to frictional forces, which results into material softening [4,5]. Ultrasonic vibrations and pressure cause diffusion in the joined materials and, as a result, their joining due to adhesion [1,2]. Importantly, ultrasonic vibrations have the additional advantage that they break the oxide layer on the surfaces of the metals to be joined [6,7]. This creates a metal-to-metal contact eventually resulting in a weld [8–10].

Due to this joining mechanism, the structure of USW welds is usually free from brittle intermetallic layers along the weld interface, which ensures a lower electrical resistance [11–13]. The USW welding technology is therefore suitable for joining metals used in electrical engineering, such as silver, copper, gold and aluminium [14].

Many parameters influence the final quality and the performance of the joint. The main process parameters are the welding time, the welding pressure, the clamping force, and the amplitude of the sonotrode [15]. The workpiece properties, such as dimensions and materials to be joined, as well as joint configuration, also play a key role on the welding process [16]. Other parameters which influence the process are the shape of the sonotrode tip [17] and the pattern on the anvil. [18,19]. Among these parameters, the thickness of the sheets is a critical factor conditioning the feasibility of the weld. Due to power limitations of the welding equipment, indeed, only thin-walled components can be successfully joined with USW [20]. The thickness limit depends on the delivered power of the machine, the tool geometry and the weldability of the materials. In addition, the ultrasound vibrations have to propagate through the upper workpiece to the weld interface. This causes the absorption of the sound wave and losses due to internal friction, with the consequent internal heat development. Therefore, the vibration amplitude of the sonotrode needs to be adjusted according to the thickness of the upper part to obtain optimal welding conditions at the interface. For thin materials, the required machine power is much lower. To weld thicker parts, on the other hand, a higher amount of energy and thus machine power is required to be able to join the workpieces. In contrast, the thickness of the bottom part is not limited as it is placed on the anvil and does not significantly affect the welding process. The thickness limit differs for different materials due to material properties, such as the hardness and their overall weldability [21–23].

According to [19], the length and width of the upper specimen are also an important factor to take into account. Different lengths and widths of the specimen will alter the resonance frequencies. This may have only little impact for parts with small dimensions but becomes more important when the sample dimensions increase. According to [20], the strength of the weld decreases when the length of the part is half of the longitudinal wavelength. In this case, the vibration is excited in a node position of the vibrating part. For larger parts, whose dimensions may be of the order of magnitude of the ultrasonic vibrational wavelengths, component resonance problems may arise. The failure load of the weld will decrease significantly at the critical length and the upper part will become heavily deformed by the sonotrode. The applied wavelength depends on the material and on the frequency used. This means that when the frequency changes, also the critical dimensions of the workpieces, at which the weld strength decreases significantly change [21,22]. The width of the top part perpendicular to the vibration direction has less influence on the weld strength. In this case, a very good weld with very little deformation of the upper part can be achieved. It points out that processing conditions need to be adapted to the specific thickness and sizes of the parts to be welded, and the influence of the parameters on weld quality and performance can change if the geometry of the parts are modified [20].

That said, the selection of the appropriate window of processing parameters to obtain a high weld quality and high strength is a challenging task for the industries interested in the application of ultrasonic welding technologies [20]. The aim of the present work is to investigate the influence of the main process parameters, namely, welding time, welding pressure and vibration amplitude, on the quality of the weld for a 0.3 mm thick copper sheets. Other parameters, mentioned above, which may influence the welding, were held constant.

## 2. Materials and Methods

The experiments were performed using sets of two copper sheets with dimensions 9.0 mm × 30.0 mm. The sheets used in the experiments had thicknesses of 0.3 mm. The sheets were welded in the overlap configuration (see Figure 1). The lower sheet is clamped to the anvil with bolts, while the second sheet is positioned freely on top of it.

EN Cu-ETP (Electrolytic Tough Pitch) copper was used for the specimens. This material is often used in electrical engineering applications because of its very good electrical conductivity. The considered material is characterised by a good toughness and plasticity,

and it is resistant to corrosion. The chemical composition of the copper plates consists of Cu and O with a minimum value of 99.90% Cu and a maximum O content of 400 ppm. The measured percentage of Cu for the plates with 0.5 mm thickness is 99.971%. The mechanical properties of the copper plates with a thickness of 0.5 mm according to the certificate (ISO 9001:2008) are given in Table 1.

**Table 1.** Nominal and measured mechanical properties of EN Cu-ETP material.

|  | Tensile Strength (Rm N/mm²) | 0.2 % Proof Strength (Rp0.2 N/mm²) | Elongation (%) | Hardness HV2 |
|---|---|---|---|---|
| Nominal | 240–300 | Min. 180 | Min. 8 | 65–95 |
| Measured | 272 | 199 | 36 | 84 |

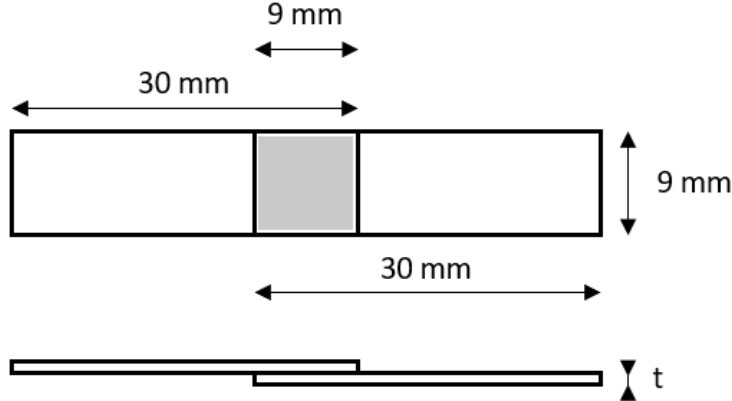

**Figure 1.** Weld configuration.

The investigations were carried out using the Telsonic MPX Ultrasonics Linear Metal Welding Press equipment. The specifications of this machine are a nominal power of 3.6 kW, a frequency of 20 kHz, the maximum load that can be applied is 1600 N and the maximum vibration amplitude is 66 µm. Using the control software, a vibration amplitude can be set between 50 and 100% of the maximum value. The sonotrode moves horizontally back and forth, resulting in tangential transmission of the vibrations into the workpieces. Below, Figure 2 shows the welding equipment employed in the present investigation.

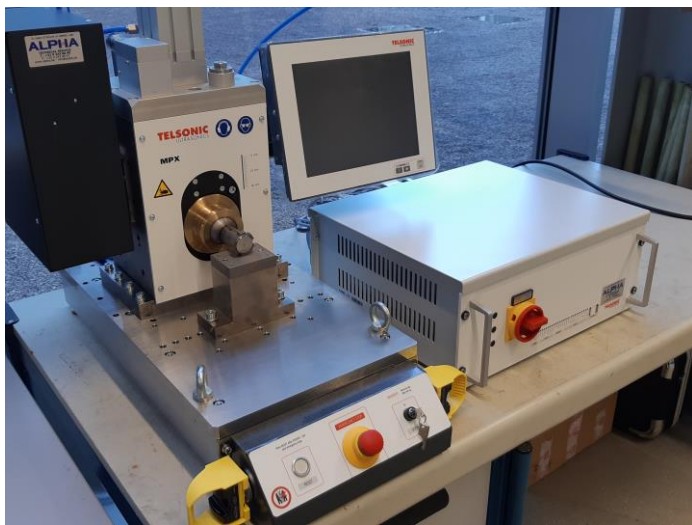

**Figure 2.** Telsonic MPX ultrasonic welding equipment.

The goal of this work was to evaluate the influence of the most important process parameters, welding time, pressure and amplitude, on the weldability of the workpieces. The influence of these three parameters, on the joint properties and weld quality was investigated lap shear testing and metallographic evaluation. Different values of these parameters were considered according to the established design of experiments (DoE). The other process parameters were kept constant for all welding tests. The trigger time is set to 0.04 s, and the pressure build-up time to 0.30 s. This means that the sonotrode starts vibrating after 0.04 s until it reaches the required pressure after 0.30 s. The holding time and pressure are chosen equal to the selected time and pressure during the weld cycle.

The workpieces were degreased using acetone. For each combination of parameters, 4 replicas were made: 1 for metallographic evaluation and 3 to perform lap shear tests to obtain a statistically relevant weld strength. The metallographic preparations were made in a plane parallel to the direction of the sonotrode vibration. Observations were made on a macroscopic scale and selected areas were subjected to additional, more detailed analysis at higher magnifications by etching the samples with a 10% ammonia solution in water saturated with hydrogen peroxide. The etching time was 20 s.

Lap shear tests were carried out using a universal testing machine. The lap shear strength of the welded specimens is measured using the Instron 8872 servo hydraulic fatigue testing system. The testing machine has a dynamic load capacity of 25 kN and uses a controlled displacement rate of 0.0333 mm/s. The tests are performed according to the EN ISO 14273 standard for tensile shear testing of spot welds in overlapping sheets. The welded specimens are clamped in the machine at both ends and pulled apart until they fail. A constant displacement speed of 2 mm/min was used during the tests. For each of the 4 samples, welded with a given parameter combination of the test plan, the welding energy was measured, while three samples were subjected to lap shear testing. Based on the obtained test results, the mean value and the standard deviation were determined. Below, a photograph of the lap shear testing machine employed is reported in Figure 3.

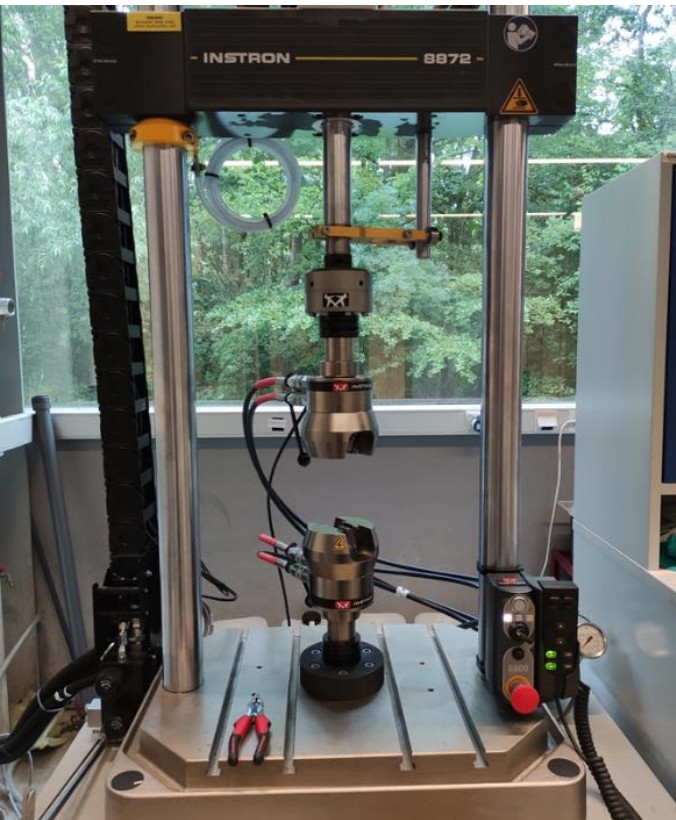

**Figure 3.** Tensile test machine Instron 8872.

To accompany the visual investigation of the welded samples, peel testing was performed. This mechanical test does not give a quantitative evaluation of the weld but is important to have an initial idea of the weld quality, in order to determine the boundary values of the process parameters. The welded samples are clamped into a vise and peeled with pliers. The goal of this test was to evaluate the appearance and location of the fracture. This test is performed manually, which makes it impossible to measure the applied force or its repeatability. The result of this test is a score of 1, 3 or 5, according to the location where the specimen breaks. A score of 1 is given to samples that are not welded or where the welded sheets can easily be separated. Samples that break at the weld interface, where a part of the welded area remains intact after peeling, are given a score of 3. Lastly, a score of 5 is given to samples that break in the base material. A visual representation of the peel test scores is shown in Figure 4. These peel test scores are based on the weld quality classifications in the work [24], where different failure modes are connected to different levels of quality.

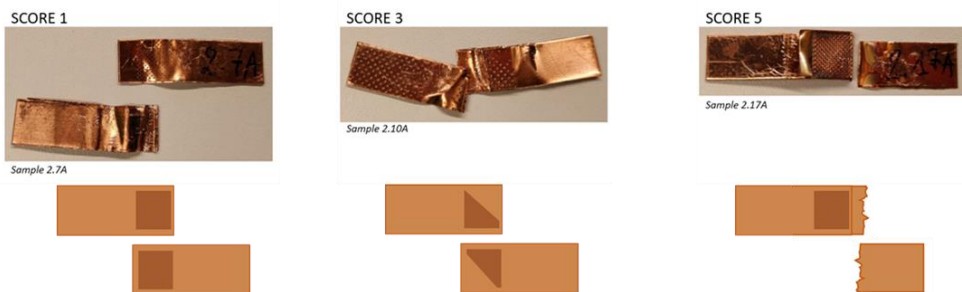

**Figure 4.** Peel test classification.

After the determination of the parameter window, a full factorial DoE was performed for each sheet thickness. The influence of the welding parameters on the welding energy and on the shear strength was then estimated.

## 3. Results

### 3.1. Prelminary Peel Test

The goal of these experiments was to evaluate the weldability window for the workpieces. A visual evaluation of each weld combined with a peel test was used to determine the boundary values of each process parameter. Too low parameter values do not create a welded joint of the workpieces. Too high parameters, however, cause some undesired phenomena during welding, including the deformation of the sheets at the weld location, bending upwards of the upper sheet during the weld cycle, and sometimes even a rupture of the sheets while clamped under the sonotrode. Figure 5 shows an example of a welded specimen that is deformed at the weld nugget and bent upwards during welding.

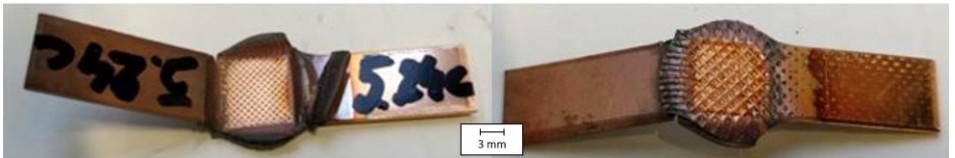

**Figure 5.** Bended and deformed weld specimen (left: top view, right: bottom view).

For the thin sheets used in this work, a large value for the pressure or vibration amplitude results in the deformation of the weld surface and even cracks in the copper sheets. A high-pressure value will result in the thinning of the workpieces, because the sonotrode will indent deeper into the material. A high vibration amplitude increases the amount of friction and heat that is introduced into the weld. For some welds, the heat development

is visible as discolorations around the weld nugget. An example of this is shown in Figure 6.

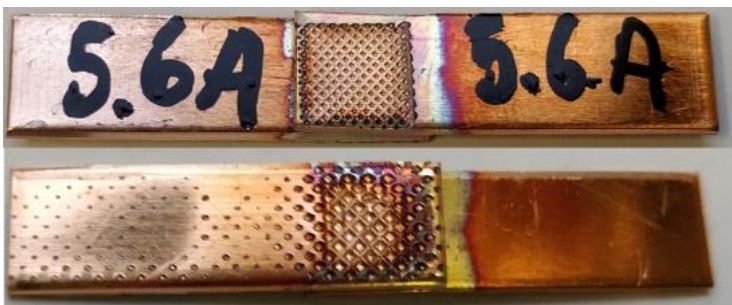

**Figure 6.** Heat development around the weld nugget (top view and bottom view, respectively).

Peel testing was performed for each parameters combination to have an initial indication of the weld quality. It is noticed that the number of welds with a peel test score of 3 or 1 was large for the 0.3 mm thick sheets. This indicates that it is difficult to achieve a satisfactory weld for a small sheet thickness of 0.3 mm, even though the parameters are chosen within the parameter window. Not all combinations of parameter values within the parameter window resulted in a sound joint between the workpieces.

The parameter settings corresponding to the different points of the DoE matrix are shown in Table 2. No intermediate value of welding time is chosen for the test series because observations during the exploratory welds indicated that the welding time had lower influence on the weld quality within the limits considered.

**Table 2.** Parameter values of the DoE matrix.

| Sheet Thickness | DoE Level | Welding Time (s) | Pressure (Bar) | Amplitude (%) |
|---|---|---|---|---|
| | −1 | 1.50 | 1.50 | 50 |
| t = 0.3 mm | 0 | / | 2.25 | 60 |
| | +1 | 3.50 | 3.00 | 70 |

According to the parameters listed in the Table 3, a full factorial plan of the experiments was designed consisting in 18 process conditions ($3^2$ combinations for the pressure and amplitude per the two values of the welding time). Four replications were made for each configuration of the process parameters, leading to a total of 72 welding tests. The following table summarizes the welding configuration and the measured welding energy consumption and lap shear strength (LSS). The energy values obtained from the data log of the machine represent the electric energy input during the weld. No information is available about the amount of energy effectively used for the weld formation, as energy losses due to heat dissipation are difficult to determine. The resulting energy response of the different welding conditions of the DoE were analysed. Each welding condition has been executed four times, resulting in four values for the welding energy. According to [25], all three welding parameters are directly linked to the energy delivered to the weld. The result from the lap shear test from the equipment is the force $F$ required to break the sample. The lap shear strength $S$ of the samples is then calculated by dividing this force by the overlapping area at the weld interface, as shown is Equation (1).

$$F = S \times A \qquad (1)$$

The areas of the cross-sections are (see Figure 1):

$$A = 9.0 \text{ mm} \times 9.0 \text{ mm} = 81 \text{ mm}^2$$

The ratio between the LSS of the welds and the calculated lap shear strength of the base material is also reported, instead of the actual values, as an indicator of the weld

quality. The base material strength was measured by performing a tensile test on three sheets with dimensions 51 mm × 9 mm × 1.0 mm, and by calculating the average of the resulting forces at failure. The length of 51 mm corresponds to the average length of the welded configurations, as each sheet is 30 mm long with an overlap of 9 mm. The average measured force to break the copper base material equals 2.175 kN. The average yield strength of the copper sheets corresponds, hence, to 241 MPa (see also Table 1). With a bonded area of 81 mm², the calculated LSS of copper sheet is approximately 23.3 MPa. The lap shear strength is a material property, making it an appropriate reference for all three sheet thicknesses.

**Table 3.** Full factorial design of experiments and the welding energy and maximum force registered in the lap shear test.

| Sample nr. | Welding Time (s) | Amplitude (%) | Pressure (bar) | Peel Test Score | Energy (J) | | Tensile Force (kN) | | | Relative LSS |
|---|---|---|---|---|---|---|---|---|---|---|
| 2.1 | 1.5 | 50 | 1.5 | 1 | 513 | ± 9.6 | 0.738 | ± | 0.021 | 1.13 |
| 2.2 | 1.5 | 50 | 2.25 | 1 | 705.6 | ± 27.2 | 0.709 | ± | 0.010 | 1.09 |
| 2.3 | 1.5 | 50 | 3 | 5 | 903.4 | ± 8.3 | 0.728 | ± | 0.009 | 1.12 |
| 2.4 | 1.5 | 60 | 1.5 | 3 | 602.1 | ± 20.6 | 0.712 | ± | 0.009 | 1.09 |
| 2.5 | 1.5 | 60 | 2.25 | 3 | 849.2 | ± 6.4 | 0.707 | ± | 0.005 | 1.08 |
| 2.6 | 1.5 | 60 | 3 | 5 | 1019.1 | ± 15.3 | 0.729 | ± | 0.010 | 1.12 |
| 2.7 | 1.5 | 70 | 1.5 | 1 | 646.1 | ± 24.7 | 0.657 | ± | 0.037 | 1.01 |
| 2.8 | 1.5 | 70 | 2.25 | 3 | 977 | ± 12.2 | 0.675 | ± | 0.022 | 1.04 |
| 2.9 | 1.5 | 70 | 3 | 5 | 1103.8 | ± 19.3 | 0.696 | ± | 0.003 | 1.07 |
| 2.1 | 3.5 | 50 | 1.5 | 3 | 1187.6 | ± 59.5 | 0.703 | ± | 0.004 | 1.08 |
| 2.11 | 3.5 | 50 | 2.25 | 5 | 1669.4 | ± 19.3 | 0.705 | ± | 0.007 | 1.08 |
| 2.12 | 3.5 | 50 | 3 | 5 | 2113.9 | ± 26.0 | 0.705 | ± | 0.008 | 1.08 |
| 2.13 | 3.5 | 60 | 1.5 | 1 | 1407.8 | ± 71.3 | 0.625 | ± | 0.105 | 0.96 |
| 2.14 | 3.5 | 60 | 2.25 | 5 | 2012.5 | ± 10.8 | 0.685 | ± | 0.014 | 1.05 |
| 2.15 | 3.5 | 60 | 3 | 5 | 2352.8 | ± 63.4 | 0.601 | ± | 0.027 | 0.92 |
| 2.16 | 3.5 | 70 | 1.5 | 5 | 1728.6 | ± 50.6 | 0.568 | ± | 0.116 | 0.87 |
| 2.17 | 3.5 | 70 | 2.25 | 5 | 2257.4 | ± 19.1 | 0.648 | ± | 0.008 | 0.99 |
| 2.18 | 3.5 | 70 | 3 | 5 | 2508.8 | ± 86.9 | 0.511 | ± | 0.091 | 0.78 |

The reported values of welding energy and force are averaged on four and three measurements, respectively. The relative lap shear strength is calculated from the averaged LSS.

### 3.2. Metallographic Examination

All welds with a peel test score of 3 or 5 have been evaluated using optical microscopy. Regardless of the welding configuration, no continuous weld interface has been observed; indeed, during the weld cycle, local welded spots are created, referred as welded islands, interrupted by interfacial gaps [26]. These welded islands are clearly visible at the weld interface (Figure 7b). On the locations between the welded islands, where no clear gaps are visible, the workpieces can be assumed to be connected to each other.

On the cross section of the welds, black lines can be also distinguished at the interface (shown in Figure 7a,b). These lines indicate gaps between the welded sheets where no bonding has occurred. Apart from these gaps and porosities, no other weld flaws have been detected in the welds.

On the images of the etched samples, very thin lines can be also distinguished in between the welded islands. This phenomenon can be seen on almost all investigated samples. These lines are not thick enough to clearly be categorized as a gap, therefore they represent locations where the sheets are joined. However, compared to the welded

islands, these lines present different features and, probably, no similar mixing of the material has occurred. This has been shown by a SEM investigation (Figure 8 and Figure 9).

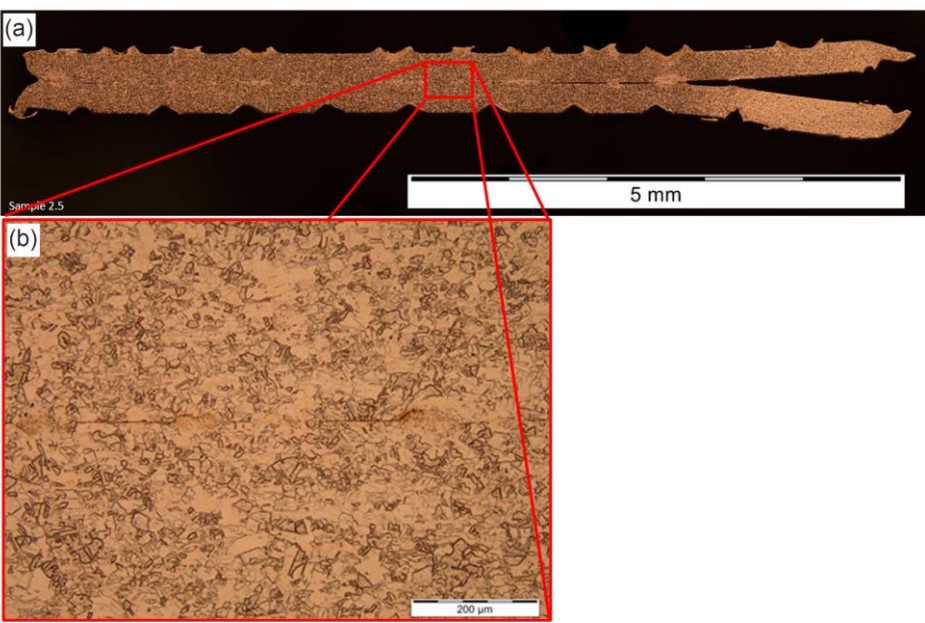

**Figure 7.** Metallographic images of: (**a**) etched Cu—Cu weld sample (×50); (**b**) detail images of the weld interface (×200).

SEM image (Figure 8a) indicate that these thin locations near the welded area are not weld flaws and the sheet are bonded without defects, despite the absence of mixing between the copper sheets. They probably represent location where diffusion bonding occurred. In this case, the weld length (as the fraction of the weld interface length where the bond has been achieved) can be calculated more easily as the size of the planar weld flaws (black lines) are much easier to determine.

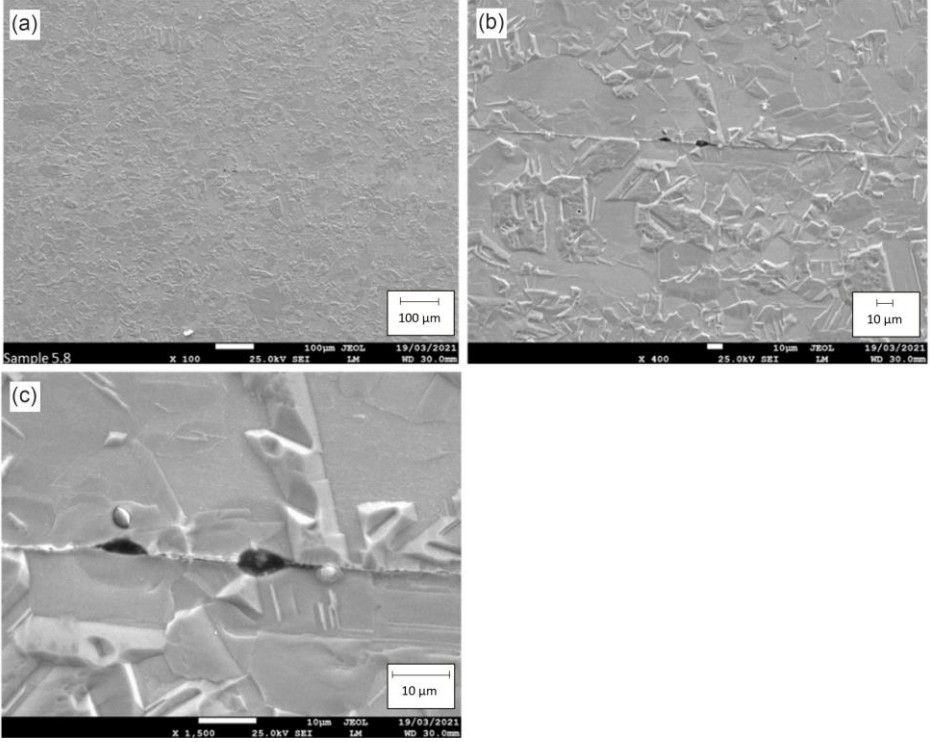

**Figure 8.** SEM image of porosities at (**a**) ×100; (**b**) ×400; (**c**) ×1500.

A closer examination of the weld interface revealed the presence of very small porosities at the weld interface. Figures 8 and 9 show these porosities at different magnification. The size of these porosities is in the order of few tens of micro-meters. From this SEM evaluation, the weld quality of this ultrasonic weld is very good as only a few very small weld flaws have been detected along the entire weld length.

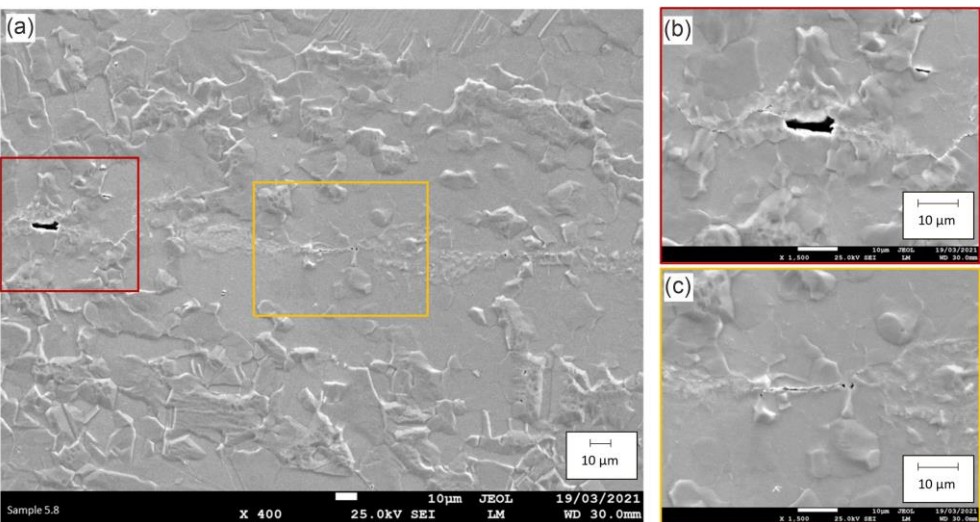

**Figure 9.** SEM image of porosities at (**a**) ×400; (**b**) and (**c**) ×1500.

A final investigation of the weld interface has been conducted by using an EDX mapping (Figure 10). The elemental maps show the presence of only Cu and the absence of a significant amount of oxygen at the weld interface; the oxygen maps, indeed, indicate a very low intensity of oxygen spectrum, and no visible concentration of the element was registered (even in the zones where porosities where detected, see Figure 10). This points out that the formation of copper oxide layer in the interface between the two mating surfaces was prevented, confirming that there is no gap present in between the welded sheets, and the line that is visible on the etched images is no weld flaw. This line is the location where both sheets are pressed together, and a bond is created.

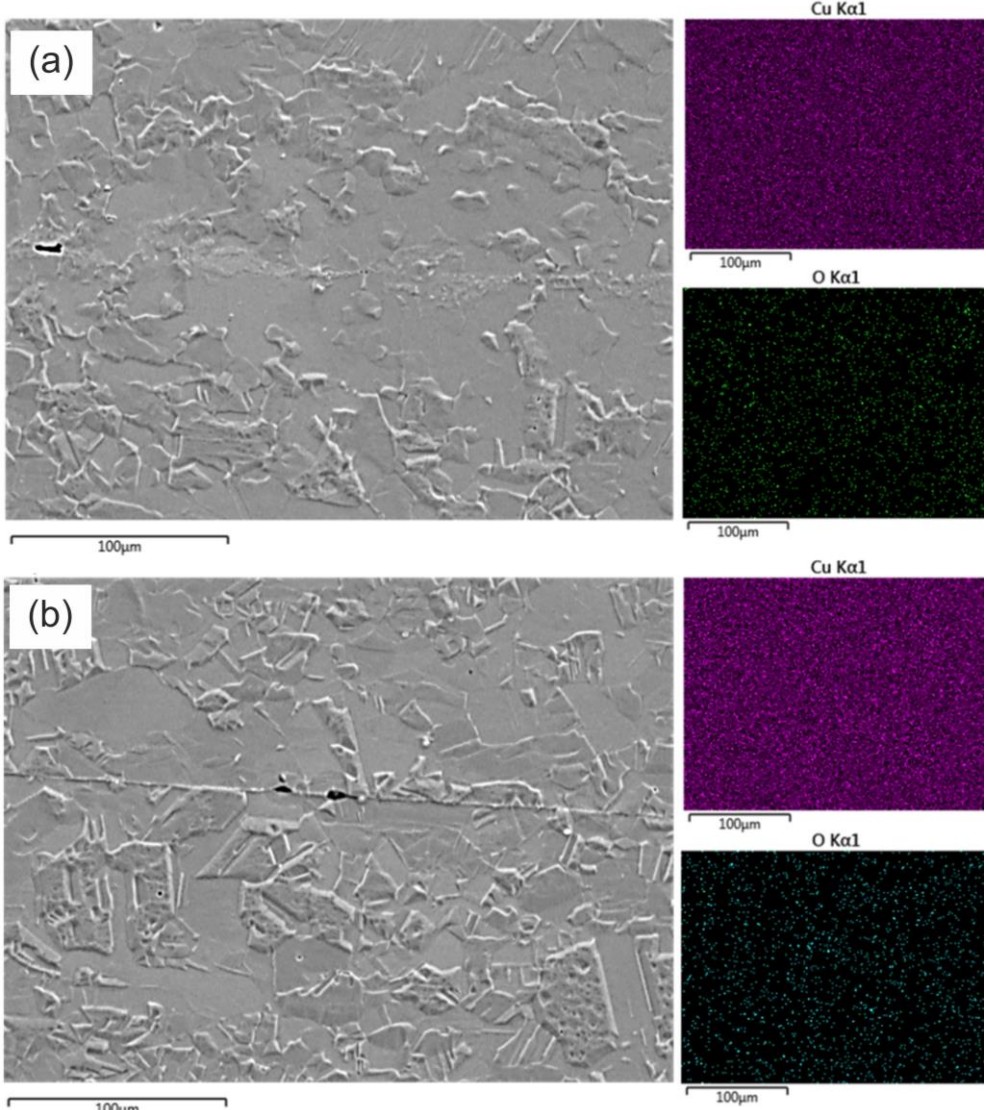

**Figure 10.** EDX mapping of Cu and O at the weld interface in two different locations along the bonding line, from: (**a**) Figure 9a; and (**b**) Figure 8b.

*3.3. Influence of the Welding Parameters on the Welding Energy and Lap Shear Strenght*

An important output parameter of the ultrasonic welding process is the energy. This value is generated by the ultrasonic welding equipment after each weld cycle. The energy value obviously differs for each welding condition based on the power absorbed during the welding.

The pareto chart of the standardized effects of all parameters and their interactions is drawn for the average of those four welding energy values of each configuration. In Figure 11, the pareto chart with the average welding energy as a response is given.

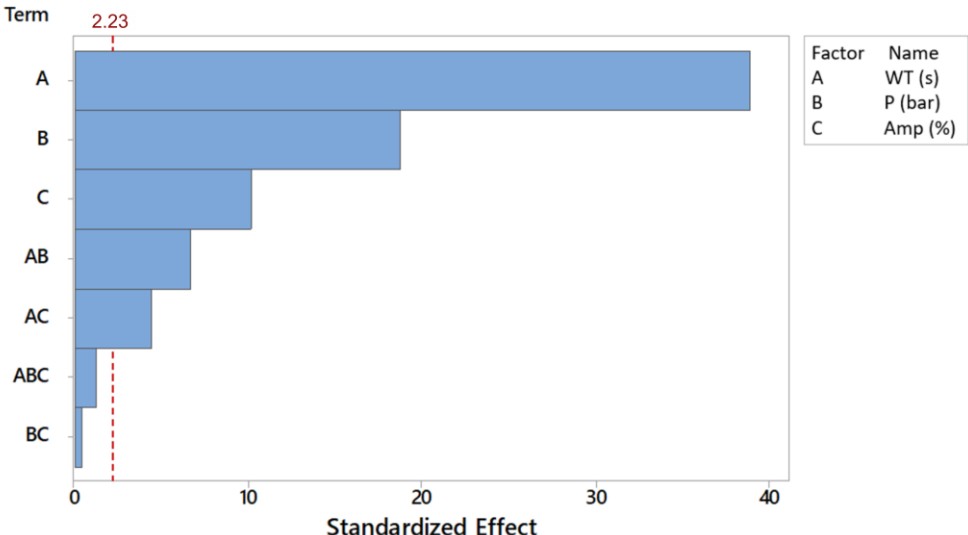

**Figure 11.** Pareto chart of the standardized effects of the welding parameters on the welding energy for a significance level of 95%.

From the energy values acquired during the experimentation and the analysis of variance the variables coefficient were calculated and used to develop the regression model of the welding energy estimated. The equation corresponding to these results is shown in below:

$$E = 982 - 735A - 405B - 18.5C + 367AB + 15.66AC + 7.58BC - 3.56ABC \quad (2)$$

with $E$ the average energy, $A$ the welding time, $B$ the pressure and $C$ the amplitude. The terms $AB$, $BC$, $AC$ and $ABC$ indicate the two-factor and the three-factor interaction between parameters. The coefficients are derived from the main effects of the parameters; the coefficient is half of the parameter effect, and its sign indicates the direction of the relationship between the term and the response. This equation shows the relative influence of each welding parameter and the interactions between the parameters on the welding energy. From this analysis it follows that the welding energy is mostly influenced by the welding time, followed by the pressure and vibration amplitude. The interactions between the parameters have less influence. However, the interactions between welding time and pressure ($AB$), and between welding time and vibration amplitude ($AC$) are statistically significant as their value is above the limit of 2.23 (for $\alpha = 0.05$). Contour plots of the average energy versus the amplitude and pressure for a constant welding time reveal an increase in energy when one of the welding parameters is increased. In Figure 12, the contour plots are shown for the minimum and maximum value of the welding time. These plots show a maximum energy value in the upper right corner of the contour plot, corresponding to the maximum value of the amplitude and pressure. The relatively straight lines indicate the limited mutual interaction between amplitude and pressure ($BC$ effect), as also highlighted by the Pareto chart. The energy values increase significantly as the welding time is increased to its maximum value (Figure 12b).

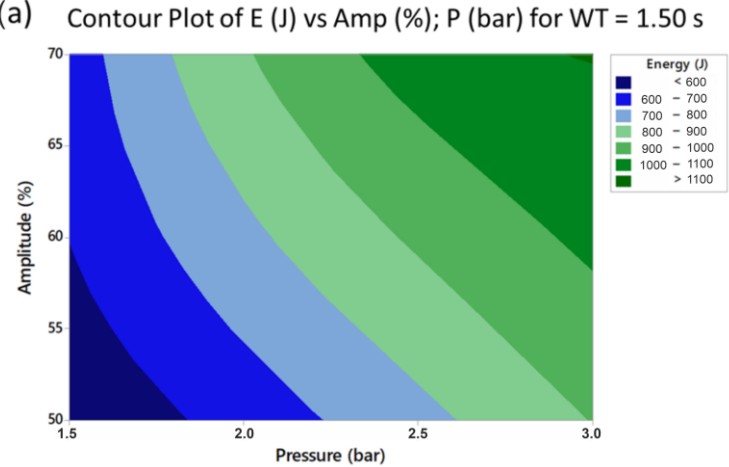

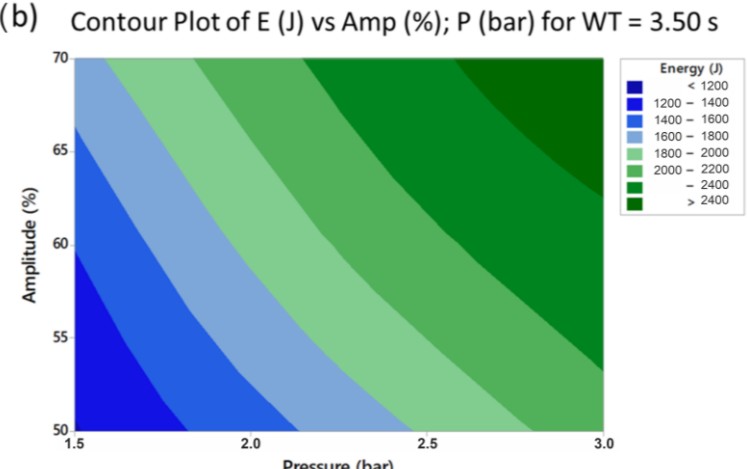

**Figure 12.** Contour plots of energy vs. amplitude and pressure at constant welding time: (**a**) 1.5 s; (**b**) 3.5 s.

The lap shear strength represents the second response of the DoE in the present work. Like the welding energy, the process parameters influenced on the lap shear strength of the welds. The lap shear strength is an indicator for the weld quality; a quality weld will fail in the base material before the weld nugget fails. For each welding condition, three replicas were subjected to lap shear testing. During welding with high values of the welding parameters, some deformations of the welded area took place. These plastic deformations are a consequence of the high heat input and pressure during the weld cycle. The application of such conditions can alter the mechanical properties of the material and weld. It is therefore important to keep these deformations in mind when analysing the lap shear strength of the welds.

Similar to the analysis of the welding energy, the influence of the welding parameters on the lap shear strength can again be analysed using a pareto chart of the standardized effects of the parameters on the average lap shear strength. The chart is shown in Figure 13.

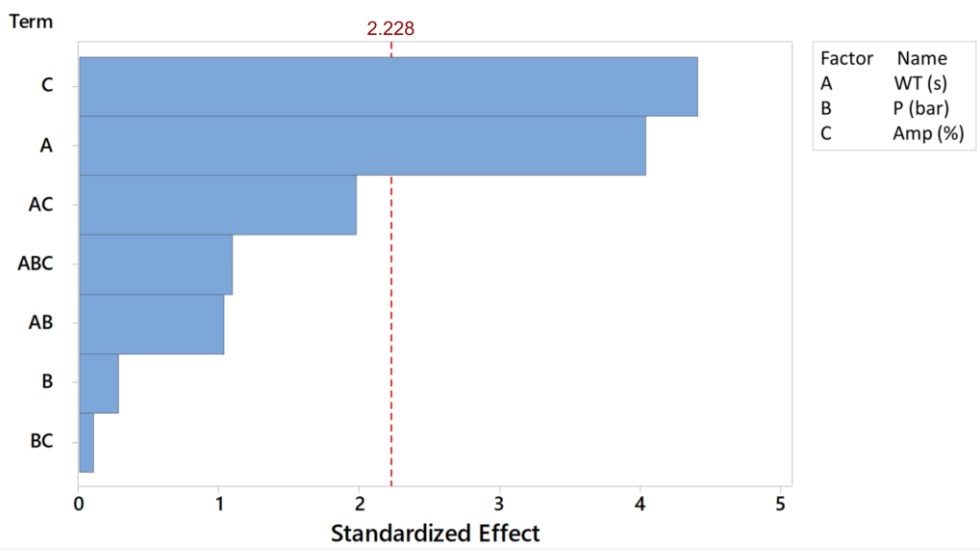

**Figure 13.** Pareto chart of the standardized effects of the welding parameters on the lap shear for a significance level of 95%.

The corresponding factor coefficients and the regression equation are reported below:

$$S = 0.459 - 0.0467A - 0.0850B - 0.00342C + 0.0350AB + 0.00077AC \\ + 0.00161BC - 0.000669ABC \tag{3}$$

with $S$ the lap shear strength, $A$ the welding time, $B$ the pressure and $C$ the vibration amplitude. By comparing the standardized effects of the two process responses it is possible to observe that the effects of the parameters on the lap shear strength are smaller than on the welding energy. In addition, differently from the welding energy, the amplitude registered the highest effect on the shear strength followed by the welding time. These are the only parameters having a statistically significant effect. An increase in pressure does not have a remarkable influence on the strength of these welds. In addition, the factors interactions are below the significance level. In the following results, instead of using the actual lap shear strength values, the weld quality is represented as a percentage of the lap shear strength on that of the base material.

The relation between the welding parameters and the lap shear strength can be visualised using contour plots (see Figure 14). These plots show the relative lap shear strength as a function of the pressure and vibration amplitude for a constant value of the welding time. Figure 14 shows that a low vibration amplitude, below the 60% of the maximum rated amplitude, results in stronger welds. The higher relevance of this parameter is highlighted by the sharp variation in the shear strength moving along the vertical axis at fixed pressure. It is worth noting that increasing the welding time, the maximum values of strength reduced (the weld/base material strength ratio changed from 1.12 to 1.05) and the shape of the contour lines changed as well, identifying different regions where high strength can be achieved. This behaviour points out that the welding time has influence on the weld strength not only per se, but also through its interaction with amplitude and pressure ($AB$ and $AC$ effects, respectively, in Figure 13). A low value for the vibration amplitude results in a higher strength. It can be observed in both shortest and longest welds. This can be explained by the higher heat input at higher vibration amplitude. This heat is the result of the increased frictional heat generation at high vibration amplitude and will cause a decrease of the strength of the workpieces. A short welding time will require a high pressure to obtain a strong weld, while for a longer welding time a lower pressure can already result in a sufficiently strong weld. For the latter condition, a high pressure can still be efficiently adopted, but lowest levels of amplitude must be chosen to balance the heat generation.

The analysis of the cross-section of samples (Figures 13–15) allows to better understand the influence of the welding variables on the effective weld length, which is connected to the strength of the joint. Increasing only the welding time, while the other parameters are kept constant, resulted in a slight increase of the welded area; the size of welded islands have grown, however, the amount of unwelded zones has not significantly decreased. The energy input of the sample shown in Figure 15a is less than half of that shown in Figure 15b, according to the shorter weld. However, the strength and the size of the welded area do not differ significantly. It points out that the shear strength is determined not only by the fraction of welded area at the weld interface, but also the local phenomena, such thinning of the cross section, material softening, etc.

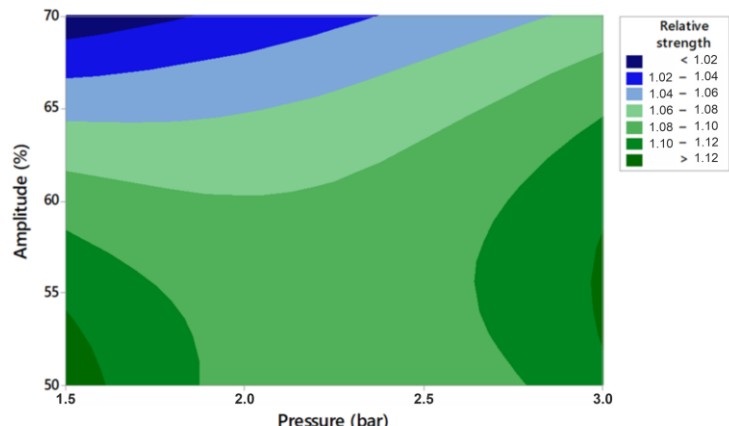

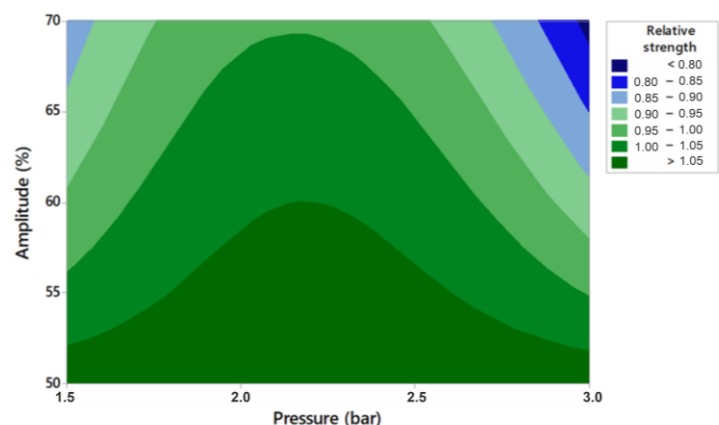

**Figure 14.** Contour plots of relative average lap shear strength vs. amplitude and pressure at constant welding time: (**a**) 1.5 s; (**b**) 3.5 s.

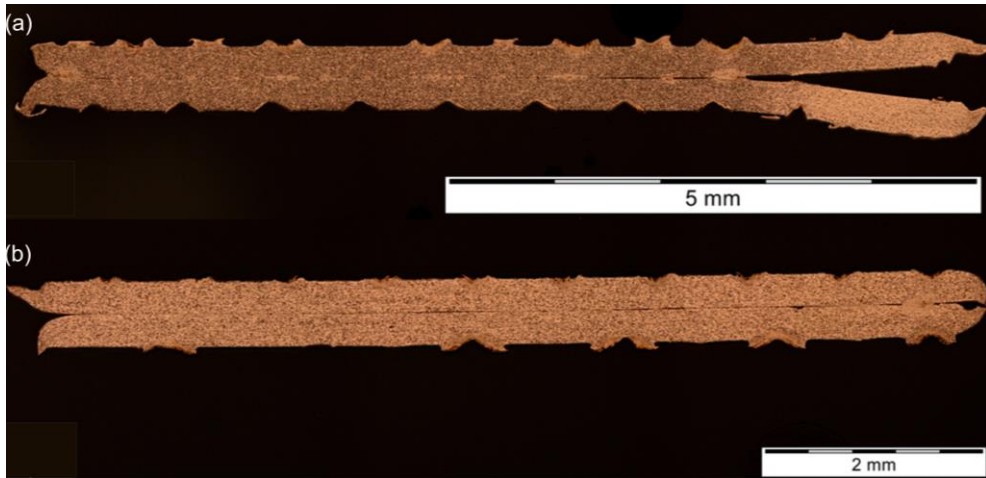

**Figure 15.** Metallographic images (×50) of etched Cu–Cu welds with t = 0.3 mm for increasing welding time: (**a**) 1.5 s; (**b**) 3.5 s.

An increase of the pressure is expected to lead to significant increase in the welded area especially in shorter welds (see Figure 14a); a higher pressure, indeed, creates suitable conditions for welded islands to form and grow (Figure 16a shows a weld with barely connected sheets and only a small welded area, while Figure 16b shows a weld interface consisting of many welded islands with small gaps in between). The relative strengths of both welds are, however, equal to 1; the amount of welded area does not change significantly, the welded islands are bigger but in between thin gaps are still present, which can trigger the failure of the joint under shear load. The impression of the sonotrode on the external side of the upper sheet is much deeper for a higher pressure, accommodating the higher closing force, despite the surface quality is still acceptable.

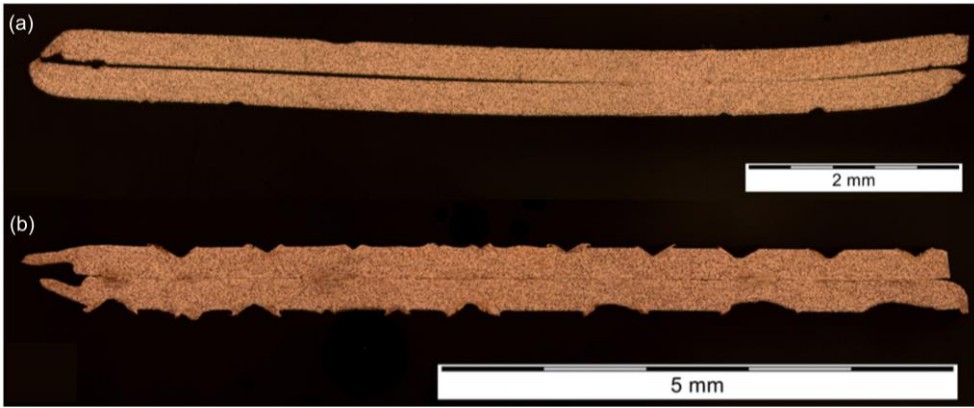

**Figure 16.** Metallographic images (×50) of etched Cu–Cu welds with t = 0.3 mm for increasing pressure: (**a**) 1.5 bar; (**b**) 3.00 bar.

Finally, Figure 17 shows welds with increasing amplitude, at fixed welding time and pressure. The amplitude seems to have reduced effect on the number of welded islands and their size. However, the length of the gaps between the sheets receded, having a potential beneficial effect on the weld strength. The amplitude, indeed, affects the amount of plastic deformation at the interface which increases with the increasing amplitude. However, too high amplitude can lead to larger deformation and thus have a negative effect on the weld strength (as observed in the contour plots at Figure 14). The sonotrode impression, also, did not changed significantly and no clear deformations of the weld nugget have occurred.

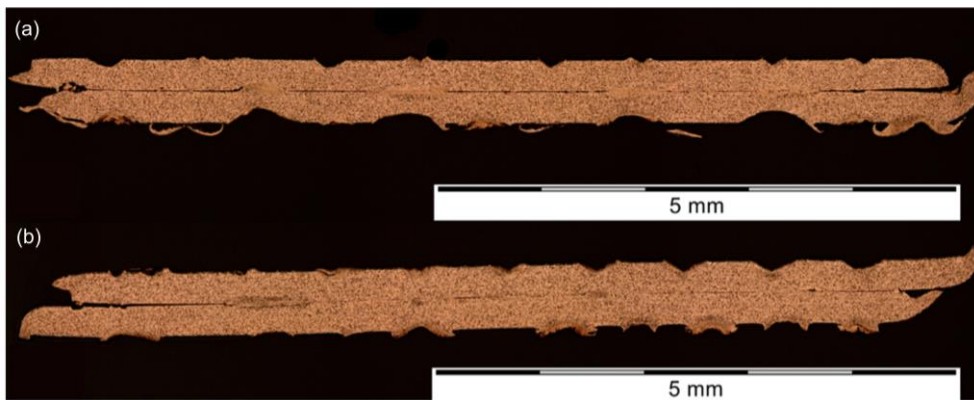

**Figure 17.** Metallographic images (×50) of etched Cu–Cu welds with t = 0.3 mm for increasing amplitude: (**a**) 50%; (**b**) 70%.

### 3.4. Relation Welding Energy—Lap Shear Strength

The welding parameters proved to have a big influence on the energy input that is delivered to the weld and on the formation of an effective bond interface, which in turn is directly correlated to the strength of the joints. Therefore, a correlation is established between these two responses. The lap shear strength of the joints is expected to increase with increasing welding energy and depends on the development of plastic deformation at the weld interface [27,28]. The plastic deformation of the surfaces enhances the amount of interfacial friction and eventually results in a larger welding area. However, with a large welding energy, a large amount of heat is introduced in the weld, causing a softening of the material and, as a result, a thinning of the sheets or even the formation of cracks around the weld nugget. These phenomena will influence the strength of the weld. The maximal weld strength can be expected at an optimized value of the input energy, where enough energy is provided to create a weld, but not too much energy so that the weld strength is compromised by crack formation or extensive thinning of the sheets.

These hypotheses are confirmed by the plot of the average energy versus the joint lap shear/base material strength ratio, reported in Figure 18, which indicates that the weld strength decreases significantly for high values of the welding energy (above 1500 J).

The majority of the welds has a weld strength higher than the base material strength. However, some of the welded specimens have a weld strength that is lower than the base material strength. Because, during the tensile tests, all samples failed in the copper base material it is possible to argue that the material properties of the base material have decreased for these samples after the joining process. At higher energy levels, indeed, correspond an increased amount of heat input. This high amount of heat will alter the mechanical properties and result in a decrease of hardness and strength. This was observed in [29]: it was found that the joint strength correlates with the temperature developed at the interface.

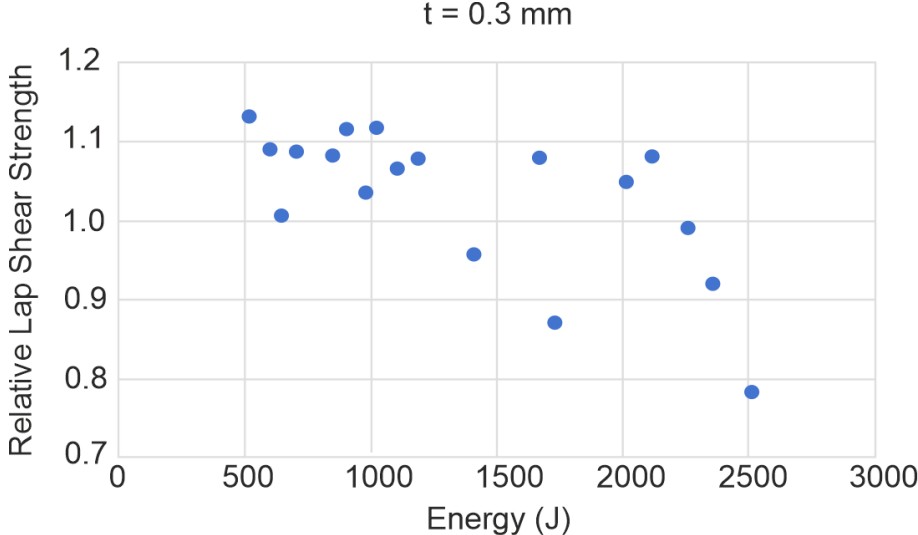

**Figure 18.** Lap shear strength vs. energy for t = 0.3 mm.

Obviously, the longer the duration of the welding process the higher is the energy dissipated by the USW equipment. Therefore, an optimal value of the welding time needs to be found for which a maximal amount of welded area can be reached. At the same time, the value of the welding time must be long enough to create a weld, but not too long to avoid that the weld quality decreases again. Moreover, an increase in energy can be the result of an increased welding pressure. A high pressure will result in thinning of the copper sheets and can change the mechanical properties of the material as well. As excepted, vibration amplitude has a direct effect on the welding energy (despite the interaction with time and pressure cannot be neglected); the increase in the entity of the relative motion between the faying surfaces inevitably lead to a higher heat generation at the weld interface, translating into the higher absorbed energy. However, the influence on the shear strength is not consistent, and the way the LSS varies with the amplitude is affected by the values of the other parameters. Overall, there is quite some scatter in the graphs, showing no clear relation between the energy and the lap shear strength. This proves that in order to achieve the maximal weld strength, an optimization of the welding parameters is necessary each time the welding conditions or weld configuration are altered.

These observations prove that an increase of any of the parameters, corresponding with an increase of the welding energy, will generally result in a larger size of the welded area. The welded islands are most often located just below the indentation of the sonotrode, as also stated in [29]. These are the locations of maximum plastic deformation due to larger stress concentrations. The effects of an increase of each of the parameters on the weld interface are summarized in Table 4.

**Table 4.** Effects of increasing parameter values on the weld interface.

| Welding Time | Pressure | Amplitude |
|---|---|---|
| - Increased welded area and size of the welded islands up to an optimal WT value | - Increase in size and number of welded islands | - Deformation, thinning and elongation of the weld nugget |
| - Deformation, thinning and elongation of the weld nugget | - Deeper sonotrode impression | - Wavy surface |
| - Wavy surface | | |

### 4. Conclusions

The goal of this analysis was to evaluate the influence of the most important process parameters on the weldability of the Cu-Cu ultrasonic welds. The weld quality is examined based on the lap shear strength and metallographic examination. The parameters window was obtained from exploratory experiments. The range of welding times that can be used to achieve a weld is quite large. The pressure and vibration amplitude are limited for the thin sheets used to avoid cracks and deformations of the welds.

The metallographic examination of the welds reveals local welded spots rather than a continuous weld formation at the weld interface. Moreover, an EDX mapping reveals that no oxygen or oxide layer is present at the weld interface. At the welded islands, bonding has taken place between the sheets. Very small porosities have been found at the weld interface. Other weld flaws have not been observed.

The welding energy is influenced by all three welding parameters and increases when one of the parameters is increased. The welding time has the biggest effect on the welding energy compared to amplitude and pressure. The pareto charts show that also the interactions between the parameters have a significant influence on the welding energy. It is therefore important that the combination of parameters is optimized rather than optimizing each parameter separately.

The lap shear strength can be used as an indicator for the weld quality; a qualitative weld will fail in the base material. A too high heat input results in the deformation of the weld interface, which in its turn results in decreased mechanical properties, such as the strength. The welding parameters are influencing the value of the lap shear strength; however, the influence of the parameters is lower than for the welding energy. The welding time has the biggest effect on the average strength but again the other parameters and the combined effects are significant and should be considered when striving for the maximal possible weld strength. The lap shear strength decreases for a high energy input, due to severe changes of the mechanical properties of the material during welding. A high vibration amplitude should be avoided as this will result in more heat input.

An optimal value of the welding time can be used to obtain the largest possible weld length. Increasing the welding time will increase the size of the welded islands, but too long a welding time results in severe deformations of the weld and a smaller welded area. These deformations translate into the thinning and elongation of the welded specimens together with the formation of a wavy pattern along the weld interface. A high pressure will result in a deeper penetration of the sonotrode inside the upper sheet and will promote the growth of the welded islands. A very high amplitude will increase the deformation of the weld. Welded islands are most often located underneath an impression of the sonotrode, at locations of maximum plastic deformation caused by large stress concentrations.

**Author Contributions:** conceptualization, S.D.M. and R.N.; methodology, K.F. and W.D.W.; validation, K.F., P.C. and F.R.; formal analysis, W.D.W.; investigation, S.D.M.; writing—original draft preparation, R.N. and K.F.; writing—review and editing, P.C., F.R. and W.D.W.; supervision, P.C. and F.R. All authors have read and agreed to the published version of the manuscript.

**Funding:** This research received no external funding.

**Data Availability Statement:** The data presented in this study are available on request from the corresponding author. The data are not publicly available; the investigation concerns internal research at the Belgian Welding Institute.

**Conflicts of Interest:** The authors declare no conflicts of interest.

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
