# Peer review of "Influence of the Process Parameters on the Properties of Cu-Cu Ultrasonic Welds"

_jmmp, doi:10.3390/jmmp7010019_

Round 1

Reviewer 1 Report

The aim of the paper was to investigate the influence of the main process parameters - welding time, welding pressure and vibration amplitude, on the welding quality for the ultrasonic welding of copper sheets.

The researches carried out for this purpose are thorough and the results are interesting.

However, I would have suggested to clarify the statement in lines 89-90 in order to bring it in agreement with e.g. lines 114-115, so maybe it would have been better to say "The experiments were performed using sets of two copper sheets ...". Also, 3 replicas for each set of parameters seems to be a really minimal amount to create a statistically relevant result.

Author Response

See attachment for the reply to the reviewer comments.

Reviewer 2 Report

In this paper, the influence of ultrasonic welding parameters on tensile strength of copper alloy sheet welding is investigated. However, the innovation of this paper is insufficient whether from the perspective of the novelty of materials or from the welding process. In addition, this paper lack of the study on the influence of process parameters on joint microstructure or fracture morphology. The experimental results are lack of solid theoretical support. In addition, the content is inconsistent with the title and the quality of the pictures is poor. The analysis of experimental results lacks references to existing literature, which means that the theoretical analysis of experimental results is not deep enough. To sum up, my recommendation is Reject. Other suggestions for this article are listed below:

1. The abstract does not correspond to the title of this paper. The title embodies the influence of lamella thickness on the joint properties; however, the abstract didn’t mention of it.

2. Some important information is missing, including chemical composition and mechanical property of the copper alloy base metal.

3. The experimental procedure is not standardized. Even though the purpose of peel test is to preliminary evaluate the appearance and location of the fracture, the process of manual execution cannot ensure the uniformity of direction and magnitude of the peel force, which may lead to unreliable results.

4. The picture quality is poor and information is incomplete. For example, Figure 3 and Figure 4 are lack of rulers, which makes the reader unable to acquire first-hand information of sample size. Figure 5 (a) is unnecessary. The ruler of Figure 6 is too vague. The contrast of Figure 8 is too low, making it difficult to distinguish the distribution of elements.

Author Response

(The authors gave the same response as above.)

Reviewer 3 Report

The paper investigated the influence of the process variables on the characteristics of the joints and weld strength in Ultrasonic welding. It's an important research direction. However, the presentation of the research is not good enough. The manuscript must be improved before it can be published.

1. In the first paragraph, reference numbers are non-consecutive as [14] is before [6,7].

2. Give a Figure of the welding equipment so that readers may have a clear understanding.

3. Give an introduction of the machine to do the lap shear testing.

4. Give a detailed introduction of the full factorial DoE. Explain how to determine the DoE levels in Table 1 and why use them.

5. How many tests are done totally? Give detailed introduction of all the test conditions and their results.

6. Give a description of a and b in Fig. 8. The mapping of O is not clear.

7. What do the standardized effects mean in Fig.9? How are they obtained? The specified values of A B C are required.

8. A is both the welding time and the area. It is not proper.

9. “From the energy values acquired during the experimentation and the analysis of variance the variables coefficient were calculated and used to develop the regression model of the welding energy estimated.” Here, how the energy values are acquired during the experimentation. There is no experimental data shown in the paper.

Author Response

(The authors gave the same response as above.)

Round 2

Reviewer 3 Report

As authors have modified the manuscript according  to reviewers' suggestions, it can be accepted now.